# Color Analysis of Merkel Cell Carcinoma: A Comparative Study with Cherry Angiomas, Hemangiomas, Basal Cell Carcinomas, and Squamous Cell Carcinomas

**DOI:** 10.3390/diagnostics14020230

**Published:** 2024-01-22

**Authors:** Dimitra Koumaki, Georgios Manios, Marios Papadakis, Aikaterini Doxastaki, Georgios Vasileiou Zacharopoulos, Alexander Katoulis, Andreas Manios

**Affiliations:** 1Dermatology Department, University Hospital of Heraklion, 71110 Heraklion, Greece; katerina.doxastaki@gmail.com; 2Department of Computer Science and Biomedical Informatics, University of Thessaly, 35100 Lamia, Greece; georgeamanios@gmail.com; 3Department of Surgery II, Witten/Herdecke University, Heusnerstrasse 40, 42283 Witten, Germany; marios_papadakis@yahoo.gr; 4Plastic Surgery Unit, Surgical Oncology Department, University Hospital of Heraklion, 71110 Heraklion, Greece; gzachar@hotmail.com (G.V.Z.); agmanios@gmail.com (A.M.); 52nd Department of Dermatology and Venereology, “Attikon” General University Hospital, Medical School, National and Kapodistrian University of Athens, Rimini 1, Haidari, 12462 Athens, Greece; alexanderkatoulis@yahoo.co.uk

**Keywords:** Merkel cell carcinoma (MCC), neuroendocrine carcinoma, skin cancer, basal cell carcinoma, squamous cell carcinoma, cherry angiomas, color analysis, image processing, computer-aided diagnostics, digital dermatology

## Abstract

Merkel cell carcinoma (MCC) is recognized as one of the most malignant skin tumors. Its rarity might explain the limited exploration of digital color studies in this area. The objective of this study was to delineate color alterations in MCCs compared to benign lesions resembling MCC, such as cherry angiomas and hemangiomas, along with other non-melanoma skin cancer lesions like basal cell carcinoma (BCC) and squamous cell carcinoma (SCC), utilizing computer-aided digital color analysis. This was a retrospective study where clinical images of the color of the lesion and adjacent normal skin from 11 patients with primary MCC, 11 patients with cherry angiomas, 12 patients with hemangiomas, and 12 patients with BCC/SCC (totaling 46 patients) were analyzed using the RGB (red, green, and blue) and the CIE Lab color system. The Lab color system aided in estimating the Individual Typology Angle (ITA) change in the skin, and these results are documented in this study. It was demonstrated that the estimation of color components can assist in the differential diagnosis of these types of lesions because there were significant differences in color parameters between MCC and other categories of skin lesions such as hemangiomas, common skin carcinomas, and cherry hemangiomas. Significant differences in values were observed in the blue color of RGB (*p* = 0.003) and the b* parameter of Lab color (*p* < 0.0001) of MCC versus cherry angiomas. Similarly, the mean a* value of Merkel cell carcinoma (MCC) compared to basal cell carcinoma and squamous cell carcinoma showed a statistically significant difference (*p* < 0.0001). Larger prospective studies are warranted to further validate the clinical application of these findings.

## 1. Introduction

Merkel cell carcinoma (MCC) is a highly aggressive form of neuroendocrine skin cancer, often proving fatal (Figure 1) [1]. The majority of cases are linked to the newly identified Merkel cell polyomavirus (MCPyV), while others are caused by mutations triggered by exposure to UV radiation [2]. This cancer is exceptionally rare, with an incidence of only 0.6 cases per 100,000 people per year in the US in 2009 [3]. While MCC is 40 times less common than malignant melanoma (MM), its survival rate is significantly lower [4]. Recent epidemiological data indicate that around 2500 new cases of MCC are reported each year within the European Union (EU), and approximately 1000 of these patients will succumb to the disease [5,6]. The high mortality rate can be attributed to the lack of effective standard treatments for metastatic MCC until recently [7,8,9,10].

MCC presents a diagnostic challenge due to its nonspecific clinical characteristics (Figure 2. It is frequently misdiagnosed as either a harmless skin lesion like a cyst, lipoma, cherry angioma, hemangioma, or other types of non-melanoma skin cancer, such as basal cell carcinoma or squamous cell carcinoma [1,11,12]. A study involving 195 patients diagnosed with MCC highlighted the difficulty in assessing the specificity of the clinical features of MCC [13]. The study emphasized that complete clinical data could not be obtained for all patients, limiting the ability to define specific clinical characteristics of MCC [13]. However, the study identified an acronym, “AEIOU” (asymptomatic/lack of tenderness, expanding rapidly, immune suppression, older than 50 years, and ultraviolet-exposed site on a person with fair skin), which may serve as clues in the diagnosis of MCC [13]. Furthermore, a European consensus-based interdisciplinary guideline on the diagnosis and treatment of MCC acknowledged the challenges associated with MCC diagnosis due to its non-specific clinical characteristics. The guideline emphasized the need for updated analysis and consensus-based interdisciplinary guidelines to address the non-specificity of MCC’s clinical features [14].

Additionally, a prospective cohort study involving 618 patients with MCC assessed the risk of stage-specific MCC recurrence and mortality over time since diagnosis. The study revealed a high 5-year recurrence rate for MCC and emphasized the importance of understanding the timing and type of MCC recurrences, further highlighting the challenges associated with managing MCC due to its non-specific clinical presentation [15]. These findings collectively underscore the non-specific clinical characteristics of MCC, emphasizing the challenges associated with diagnosing and managing the disease based solely on its clinical features. To avoid delays in diagnosis, a high level of suspicion is necessary. Traditional diagnostic methods often rely on visual inspection and histopathological analysis, which might not fully capture the nuanced characteristics indicative of MCC. However, recent advancements in digital imaging and color analysis techniques offer a promising avenue for improving the accuracy and reliability of MCC diagnosis, although this area has not yet been extensively explored in the context of MCC.

The unique color characteristics of MCC lesions, exhibiting a spectrum of colors such as red, purple, and pinkish hues, create hurdles for precise visual assessment [16,17,18,19,20,21,22]. To overcome this challenge, we have turned to color analysis techniques aimed at capturing and quantifying these subtle color variations, providing a more comprehensive and accurate representation of the disease. Among these techniques is the CIE Lab color model, specifically designed to approximate human vision and perception of color, rendering it particularly useful for color analysis in medical imaging. Through leveraging these color analysis techniques, researchers aim to extract meaningful color features and patterns that may contribute to the development of more effective diagnostic and analytical techniques for MCC. This approach holds significant potential to enhance the accuracy and reliability of MCC diagnosis, ultimately contributing to improved patient outcomes and better management of this rare and aggressive form of skin cancer.

While numerous studies have focused on systems for predicting and diagnosing melanocytic lesions [23,24,25,26,27,28], there is a noticeable absence of such studies concerning MCC. Thus, we conducted a study to analyze the digital color characteristics of MCC lesions compared to both benign lesions that bear resemblance to MCC, such as cherry angiomas and hemangiomas, and malignant lesions like BCC and SCC. Our analysis encompassed two of the most commonly used color systems: the RGB (red, green, and blue) and CIE Lab.

## 2. Materials and Methods

We conducted a retrospective study based on electronic health records of all patients diagnosed with MCC at a tertiary hospital in Germany over ten years. The research focused on individuals with histologically confirmed MCC, who had undergone lesion removal during the study period. Additionally, we included digital photographs of cherry angiomas, hemangiomas, BCCs, and SCCs. Clinical images that were blurred or involved skin lesions with scarring due to potential prior biopsies were excluded from the study. Clinical images, captured using a digital camera along with a handheld ruler positioned next to the lesion, were included in the study. The color of the lesion and adjacent normal skin from 11 patients with primary MCC, 11 patients with cherry angiomas, 12 patients with hemangiomas, and 12 patients with BCC/SCC (totaling 46 patients) was analyzed using two different color systems: CIE Lab and RGB.

### 2.1. Image Processing

Image processing was performed on the acquired color images using primarily non-commercial custom software (MEDIMPRO v.3.0), developed by one of the authors for research purposes [27,28]. This software incorporates various algorithms to facilitate tasks such as image segmentation, geometry assessment, and the analysis of color and color textures.

The study encompassed 21 variables, including the three mean values of RGB (red, green, and blue), the three components of the Lab color system (L*, a*, b*), and, lastly, the mean Individual Typology Angle (ITA), derived from the Lab Color. The same variables were assessed for the surrounding normal skin, and the differences between normal skin and the tumor were also analyzed. For the purpose of this presentation, we will refrain from discussing the details of RGB and CIE Lab color, as this information is readily available to readers, and we will provide only basic information.

The boundaries of the skin lesion were determined either automatically or, in cases where segmentation was challenging, by manually selecting points along the border and subsequently connecting them using a second-order spline curve, which minimized the mean square distance.

### 2.2. Color Space

A color space serves as a mathematical framework for expressing color data through three or four distinct color components. Various color spaces or models find application in diverse fields like computer graphics, image processing, and computer vision. Skin detection employs various color spaces, including the RGB-based color space (comprising RGB and normalized RGB), hue-based color space, and luminance-based color space [29].

### 2.3. RGB Color System

The RGB color space is a widely used color model that represents colors by combining red, green, and blue primary colors [29]. It is based on the additive color model, where different intensities of red, green, and blue light are combined to create a wide range of colors. Every color can be achieved by blending the three primary colors, and the resulting color depends on the proportion of each primary color used. Conversely, it is possible to deconstruct a particular color into its red, blue, and green constituents using this method in reverse.

When all three colors are combined at full intensity, they create white light. When all three primary colors are absent or at zero intensity, they create black. In the RGB color space, colors are typically represented as a three-dimensional numeric array, with each element specifying the intensity value of the red, green, and blue color channels. The range of numeric values depends on the data type of the image. Normalized RGB is a representation that can be effortlessly derived from RGB values through a straightforward normalization process.

The RGB color space is commonly used in various fields, including digital imaging, computer graphics, and display technologies. The RGB color system has been used in the diagnosis of skin cancer, particularly in the detection of malignant melanoma (MM).

### 2.4. Lab Color System

The LAB color system, also known as the CIE Lab color space, is a standardized color model employed for the objective description and quantification of colors [29].

Key points about the LAB color system include:
Dimensions: The L color space consists of three dimensions. L (lightness) represents the brightness or darkness of a color; an a* value of 0 represents black, and a value of 100 represents white. a* (redness–greenness) represents the position on the red–green axis. Positive values indicate redness, while negative values indicate greenness. b* (blueness–yellowness) represents the position on the blue–yellow axis. Positive values indicate blueness, while negative values indicate yellowness.Perceptually Uniform: The LAB color space is designed to achieve near-uniform spacing of perceived color differences. This means that a specific numerical difference in the LAB values roughly corresponds to a similar perceived difference in color.Standardization: The LAB color system was developed by the International Commission on Illumination (CIE) in 1976 as a standard for color communication. It provides a consistent and objective way to characterize colors, allowing for accurate measurement and comparison of all perceivable colors.Applications: The LAB color system is widely used in various industries, including printing, textiles, paint, and coatings. It enables color matching, quality control, and color communication between different stakeholders.Relationship to Other Color Spaces: The LAB color space is device-independent, meaning it is not tied to a specific device or medium. It can be converted from and to other color spaces, such as RGB (red, green, and blue) and CMYK (cyan, magenta, yellow, and key), allowing for seamless color management and translation between different systems.


The Lab color system has also been employed in the realm of skin cancer detection. It is used for color analysis of skin lesion regions to discriminate MM in clinical images. Image-based computer-aided diagnosis systems hold significant potential for the screening and early detection of MM, and the Lab color system is a key component in these systems.

When selecting color models for analyzing MCC, the RGB and CIE Lab color models stand out due to their distinct characteristics and advantages in representing and analyzing color information. The RGB color model is commonly chosen for its simplicity and direct correspondence to how colors are displayed on electronic screens. It portrays colors through combinations of red, green, and blue values, rendering it suitable for capturing color information from clinical images of MCC. Conversely, the CIE Lab color model is designed to approximate human vision and the perception of color, making it particularly valuable for color analysis in medical imaging. Comprising three components—L* for lightness, a* for the green–red color component, and b* for the blue–yellow color component—the L* component closely aligns with the human perception of lightness, facilitating the analysis of subtle color variations that may signify MCC. Moreover, the CIE Lab color model is recognized for its ability to represent all perceivable colors, making it an ideal choice for capturing the diverse range of colors present in MCC clinical images. Its perceptual uniformity ensures a consistent representation of color differences, which is crucial for accurate color analysis in medical imaging. In summary, the selection of RGB and CIE Lab color models for MCC analysis is driven by their capacity to accurately represent and analyze color information from clinical images, along with their aptness to capture the nuanced color variations that may indicate MCC. These models equip researchers with the necessary tools to extract meaningful color features and patterns from medical images, ultimately contributing to the development of effective diagnostic and analytical techniques for MCC.

### 2.5. Individual Typology Angle (ITA)

The ITA serves as a measure of skin pigmentation [30]. It classifies skin types into six groups, ranging from very light to dark skin [29,30], with higher ITA values indicating lighter skin. ITA is measured using spectrophotometric techniques and is considered an objective method for standardizing skin type classifications. In this article, the ITA is calculated using the CIE Lab color system primarily to demonstrate its change to normal skin caused by the MCC, rather than as a direct measure of ITA itself. The ITA is computed from the Lab color space according to Formula (1):ITA = [arctan(L* − 50)/b*)] 180/p(1)

Here, L* represents luminance, ranging from black (0) to white (100), and b* spans from yellow to blue.

### 2.6. Statistical Analysis

Basic descriptive statistics for the variables were computed using the Apache OpenOffice Calc 4.1.6 spreadsheet program. Comparisons between similar variables among different lesion categories were performed using the Mann–Whitney U test in the statistical software SPSS version 23. The results have been tabulated and will be elaborated upon below.

## 3. Results

The digital images of 12 patients with primary MCC were collected retrospectively. Out of the total 12 cases assessed for MCC, 1 case was excluded due to a prior biopsy resulting in a scar on the lesion. Therefore, among the 12 cases initially evaluated, 11 were included in the final analysis of this study. The images of the tumor MCC and adjacent healthy skin of all 11 patients were analyzed with RGB and Lab color techniques. Additionally, 11 patients with cherry angiomas, 12 patients with hemangiomas, and 12 patients with BCC/SCC (totaling 46 patients) were included in this study.

Before moving on to the results, we will describe a typical case depicted in Figure 1. The 82-year-old patient presented with a growing lesion, approximately three centimeters in diameter on the neck that had been observed three months ago. The patient’s history did not reveal any other evidence of malignancies or related diseases. The patient’s skin color, classified as phototype I, was noted at the beginning of the examination. Due to the short history and the lesion’s increasing size, it was suspected that the lesion was MCC. This was followed by a wide excision of the lesion where the initial assessment was confirmed. The patient’s recovery after surgery was uneventful, without any specific complications.

It should be emphasized that, due to the retrospective nature of this study, certain photographs, like the one in Figure 1, exhibited issues primarily related to light reflections. These reflections appeared in the center of the vaulted lesion due to the flash, which caused them with its bright light. However, it is important to note that these reflections did not significantly alter the results regarding the essence of this study, as their removal would strengthen our conclusions. As mentioned earlier, the lesion boundary is automatically determined by the software, as in this case. If the lesion boundary was indistinct and the automatic segmentation was unsatisfactory, manual delineation was performed.

The geometric features of the lesion were as follows:Maximal diameter: 3.04 cm;Surface: 6.73 cm^2^;Perimeter: 9.40 cm.

In Figure 2, the histograms depicted clearly illustrate the most significant reduction in green color, shifting the green histogram to the left.

Also shown in Figure 3 are the ITA graphs as computed by the program in both normal tissue and pathological tissue. Figure 3 also depicts the mean skin color of normal skin and MCC. ITA is a sensitive indicator of skin color change, and we believe that it has not been properly utilized by the scientific and especially the dermatological community. Although it has been used to determine the skin phototype, we think that its variation has more value because any pathological condition affects its value and could even be an objective indicator of the evolution of skin diseases and the evaluation of dermatological treatments. The regression of lesions would mean a return to normal color, while deterioration and persistence would indicate treatment failure. We believe that in future studies, lesions such as scars and rashes subjected to treatment will be followed, and the change in this angle will be evaluated. 

Figure 4 displays the categories of examined lesions, which include MCC Figure 4a, cherry angiomas Figure 4b, SCC Figure 4c, and hemangioma Figure 4d. 

In Table 1, out of 21 calculated parameters, we present 7, specifically the results for the means of red, green, blue (RGB), L*, a*, b*, and ITA across the examined lesions. The results encompass lesions, including MCC (number = 11), cherry hemangioma (number = 11), BCC, SCC (number = 12), and hemangioma (number = 12). 

Meanwhile, in Table 2, the comparison of mean values for red, green, blue (RGB), L*, a*, b*, and ITA across examined lesions of MCC, cherry angioma, BCC, SCC, and hemangioma is displayed.

Figure 5 illustrates the mean color values of red, green, and blue (RGB) as well as L*, a*, b*, and ITA for lesions across all four categories: (a) MCC is highlighted in red; (b) cherry angioma is represented in blue; (c) BCC and SCC are denoted in green; and (d) hemangioma is depicted in purple.

Statistical comparisons of red, green, blue (RGB), L*, a*, b*, and ITA across examined lesions, comparing MCC with cherry angioma, can be seen in Table 3. Significantly different values were observed in the blue color of RGB (*p* = 0.003) and the b* parameter (*p* < 0.0001) of MCC versus cherry angiomas. 

Figure 6 displays a box plot comparing the mean blue color values of RGB between MCC and cherry hemangioma, demonstrating a statistically significant difference (*p* = 0.03). 

Figure 7 exhibits a box plot comparing the mean b* color values between MCC and cherry hemangioma, indicating a statistically significant difference (*p* < 0.0001). 

In Figure 8, the box plot compares the mean green color value of RGB between MCC, BCC, and SCC, illustrating a statistically significant difference. 

Furthermore, Figure 9 showcases a box plot comparing the mean a* value of MCC and that of BCC and SCC, highlighting a statistically significant difference (*p* < 0.0001). 

Table 4 displays the statistical analysis comparing MCC with BCC and SCC. Significantly different values were observed in the green color of RGB (*p* = 0.037) and the a* parameter (*p* < 0.0001). 

Furthermore, a statistical analysis comparing MCC with hemangioma was conducted, revealing significant differences in the red color of RGB (*p* = 0.002), L* parameter (*p* = 0.019), a* parameter (*p* = 0.023), and ITA (*p* = 0.009) (Table 5). 

The box plot comparing the mean red value of MCC and hemangioma, showing a statistically significant difference (*p* = 0.002), is shown in Figure 10.

We calculated the sensitivity and specificity for distinguishing between MCC and cherry hemangioma using the b* color parameter as the test. Both sensitivity and specificity were 100% for each group within these small groups of patients. In the case of differentiating between skin cancer and MCC using the a* parameter as the test, we obtained identical results. When assessing sensitivity and specificity for distinguishing MCC and hemangiomas using the a* parameter at the threshold of 37, we obtained a sensitivity value of 82%, specificity of 66.6%, accuracy of 74%, a positive predictive value (PPV) of 69.23%, and a negative predictive value (NPV) of 80%.

## 4. Discussion

There is a paucity of clinical diagnostic methods available for identifying MCC. To the best of our knowledge, this is the first study in the color analysis of MCC. During this study, we calculated color attributes in both the RGB and the CIE Lab color model for the digital images of MCC patients in both lesional and adjacent healthy skin and also in patients with cherry angiomas, hemangiomas, BCCs, and SCCs. Using the Lab color system, we estimated the change in the ITA of the skin. What becomes evident when comparing the color changes is a notable decline in the presence of green in the tumor regions, accompanied by a decrease in the ITA angle. The decrease in the green color is more noticeable than the alterations detected in the other color elements in the RGB system. Within the Lab color model, we also note substantial shifts in the L* and a* parameters, whereas the b* parameter remains relatively consistent. When we performed statistical comparisons of red, green, blue (RGB), L*, a*, b*, and ITA across examined lesions comparing MCC with cherry hemangioma, we found a statistically significant difference in the blue color of RGB (*p* = 0.003) and the b* parameter (*p* < 0.0001). When comparing MCC with BCC and SCC, there was a statistically significant difference in the green color of RGB (*p* = 0.037) and the a* parameter (*p* < 0.0001). When comparing MCC with hemangioma, there was a statistically significant difference in the red color of RGB (*p* = 0.002), L* parameter (*p* = 0.019), a* parameter (*p* = 0.023), and ITA (*p* = 0.009). Previous CADx studies in dermatology based on digitized color images or dermatoscopic images mainly focused on MM or melanocytic skin lesions [31,32,33]. Developing a computer-aided diagnostic support system for skin cancer is a promising area of research.

Using computer-aided image analysis in dermatology has several benefits, including the following. (1) Improved accuracy and efficiency: AI’s ability to learn skin lesions’ features far exceeds that of humans, allowing it to quantify features and make judgments to assist in the discovery and analysis of lesions, improving the accuracy and efficiency of clinicians’ diagnoses [31,32,33]. Aid in diagnosis: Computer-aided diagnosis (CAD) techniques can assist doctors in enhancing their investigation. Image-based computer-aided systems have significant potential for screening. (2) Multi-classification for skin lesions: Developing a computer-aided diagnostic support system for skin cancer is a promising area of research. AI in the Aid-Diagnosis and Multi-Classification of Skin Lesions is a highly effective computer algorithm based on the analysis of 837 melanocytic lesions [34].

Artificial intelligence (AI) has shown promise in improving the accuracy of skin cancer diagnosis [35,36,37]. AI-based image classification methods have been developed to assist in the diagnosis of skin cancer. These methods use deep learning algorithms to analyze images of skin lesions and identify features that are characteristic of various types of skin cancer. AI researchers consistently assert that their systems outperform dermatologists in diagnosing skin cancer. However, this representation is markedly different from the actual situation, as these experiments occur within controlled environments governed by predetermined regulations. Given the numerous obstacles highlighted earlier, how these reported performance assessments take place bears little resemblance to the genuine diagnostic practices undertaken by clinicians treating skin cancer patients. Frequently, deep learning algorithms are labeled as opaque, as they solely derive insights from pixel data in imaging datasets, lacking any domain-specific knowledge or the ability to draw logical deductions to establish connections among diverse types of skin lesions. AI has been used in the diagnosis of skin cancer in clinical images. Clinical images are commonly taken using mobile cameras to enable remote examination and integration into patient medical records. Due to the diverse cameras used, along with varying backgrounds, lighting conditions, and colors, these images offer distinct insights from dermatoscopic images.

Yang et al. conducted a diagnosis of clinical skin lesions using an approach based on the ABCD rule with the SD-198 dataset [35]. They contrasted the performance of their proposed techniques with deep learning methods and dermatologists. Their approach achieved an accuracy score of 57.62%, surpassing the highest-performing deep learning method (ResNet) which achieved 53.35% [35]. In comparison to clinicians, only experienced senior clinicians with extensive knowledge of skin diseases achieved an average accuracy of 83.29%. In their research, Han et al. utilized a deep learning model (ResNet-152) to categorize clinical images depicting 12 different skin conditions [36]. They trained the model using an Asan training database, the MED-NODE database, and Atlas site images. Subsequently, they evaluated the model’s performance on both an Asan testing set and the Edinburgh Dataset (Dermofit). Remarkably, the algorithm’s performance closely matched that of the group of 16 dermatologists when tested on a subset of 480 randomly selected images from the Asan test dataset (260 images) and the Edinburgh Dataset (220 images). Furthermore, the AI system outperformed dermatologists in diagnosing BCC [36].

In their study, Fujisawa et al. evaluated a deep learning technique using 6009 clinical images encompassing 14 distinct diagnoses covering both malignant and benign cases [37]. The deep learning algorithm attained a diagnostic accuracy rate of 76.5%, surpassing the achievements of 13 certified dermatologists (59.7%) and 9 dermatology trainees (41.7%) who worked with a dataset containing 140 images [37].

Several studies have sought to identify specific markers for differentiating skin cancer from other benign tumors [38,39,40]. One study aimed to address the challenges in differentiating between trichoadenoma, trichofolliculoma, trichoepithelioma, trichoblastoma, and BCC, especially with small specimens [38]. The researchers examined 30 cases each of benign tumors from hair follicle appendages and BCC. CD10 expression was assessed in both tumor groups, and the results indicated stronger stromal CD10 immunopositivity in benign tumors, while peripheral CD10 immunopositivity was stronger in BCC. The study suggests that CD10 expression analysis can be a valuable tool for differential diagnosis, especially in small and superficial biopsies, potentially impacting treatment decisions in selected cases. Another study aimed to address challenges in differentiating between benign tumors of cutaneous appendages originating from hair follicles (BTCOHF) and BCCs, particularly with small biopsy specimens. The researchers investigated the potential utility of CD34 expression for assistance in this differential diagnosis. Results showed that the [1+] and [2+] immunopositivity of CD34 in BTCOHFs was significantly stronger than in BCCs, suggesting that CD34 expression analysis may contribute to the differential diagnosis of these skin lesions [39].

## 5. Conclusions

In our study, several limitations exist. The archival nature of our material and the small sample size, stemming from the rarity of this tumor, suggest caution in drawing definitive conclusions. Notably, the photographs of the lesions were not standardized using a protocol, leading to the unavailability of high-quality images. However, image processing, even with digital camera images, can aid in the differential diagnosis of these lesions through the examination of the aforementioned parameters.

Dermoscopy is expected to provide higher-quality images that can be more effectively utilized by computer-based analysis, and this approach is anticipated to yield positive results in the near future [27,28,29,30,31,32,33]. Diagnosis is primarily clinical, relying on the patient’s history and histological examination of the lesion, particularly in advanced stages of the disease. What does appear evident is a significant reduction in the mean green value within the lesions compared to normal skin. Additionally, changes in brightness (L*) and the a* component of skin color in the Lab system exhibit characteristic patterns along with a marked decrease in ITA. It is worth mentioning that the existing literature on the application of digital diagnostic methods for MCC is scarce to non-existent.

In the future, it is imperative to foster collaboration among various surgical oncology and dermatology centers to develop standardized protocols for recording and photographing such rare lesions, while also incorporating dermoscopic imaging. Additionally, exploring other lesions with similar morphology for digital diagnosis should be pursued as a viable alternative method. Moreover, AI is poised to play a significant role in dermatological diagnosis soon.

## Figures and Tables

**Figure 1 diagnostics-14-00230-f001:**
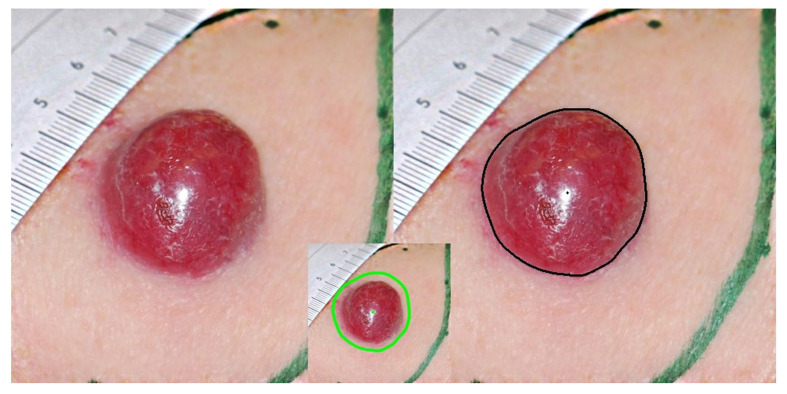
The left part of this figure is the photograph of the lesion, and the right part is the same photograph showing the boundary that was identified. In the small window in the center, there is the area of surrounding normal skin, which is obtained by incrementing the border of the lesion by 25% and taking a pixel band to estimate the parameters of normal skin depicted in the table along with the geometric characteristics. This zone in this patient corresponds to the area with the light green color.

**Figure 2 diagnostics-14-00230-f002:**
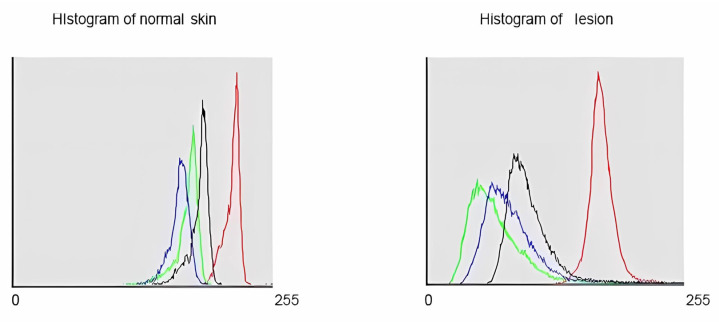
Histograms representing the color components of both healthy (**left**) and affected tissue (**right**), revealing a significant decrease in green within the lesion.

**Figure 3 diagnostics-14-00230-f003:**
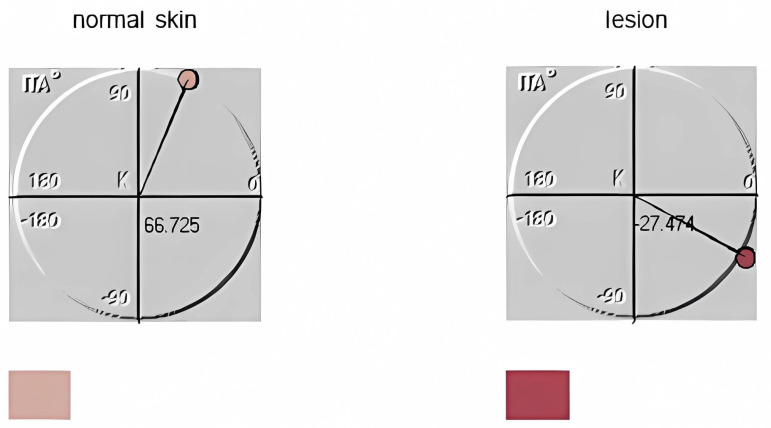
Additionally displayed in this figure are the ITA graphs generated by the software for both healthy and pathological tissues. This figure also includes representations of the average skin color for normal skin and MCC in the small boxes located beneath the ITA graph.

**Figure 4 diagnostics-14-00230-f004:**
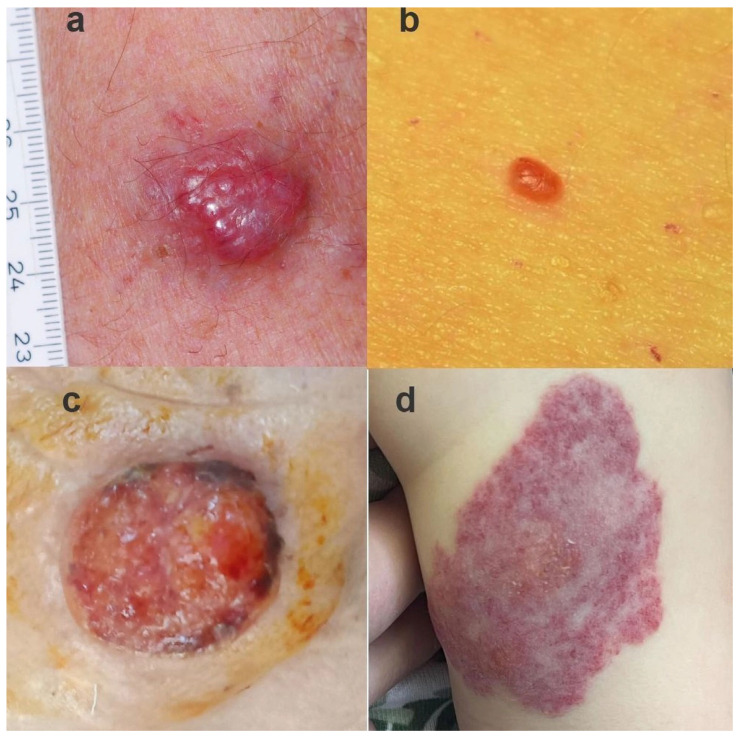
(**a**) MCC in a patient; (**b**) cherry angioma; (**c**) squamous cell carcinoma; and (**d**) hemangioma.

**Figure 5 diagnostics-14-00230-f005:**
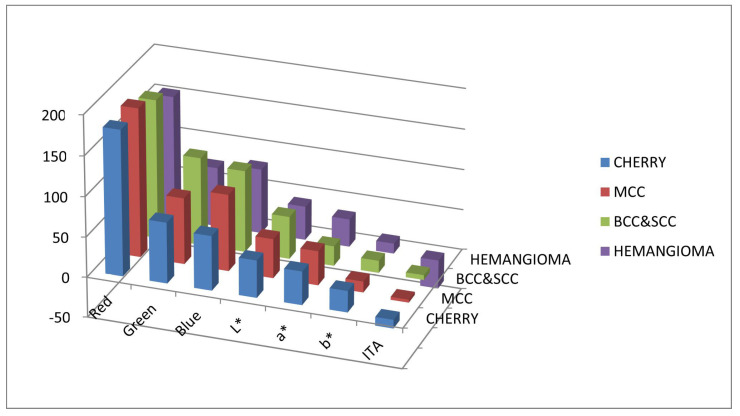
This figure illustrates the mean color values of red, green, and blue (RGB) as well as L*, a*, b*, and ITA for lesions across all four categories: (a) MCC is highlighted in red; (b) cherry angioma is represented in blue; (c) BCC and SCC are denoted in green; and (d) hemangioma is depicted in purple.

**Figure 6 diagnostics-14-00230-f006:**
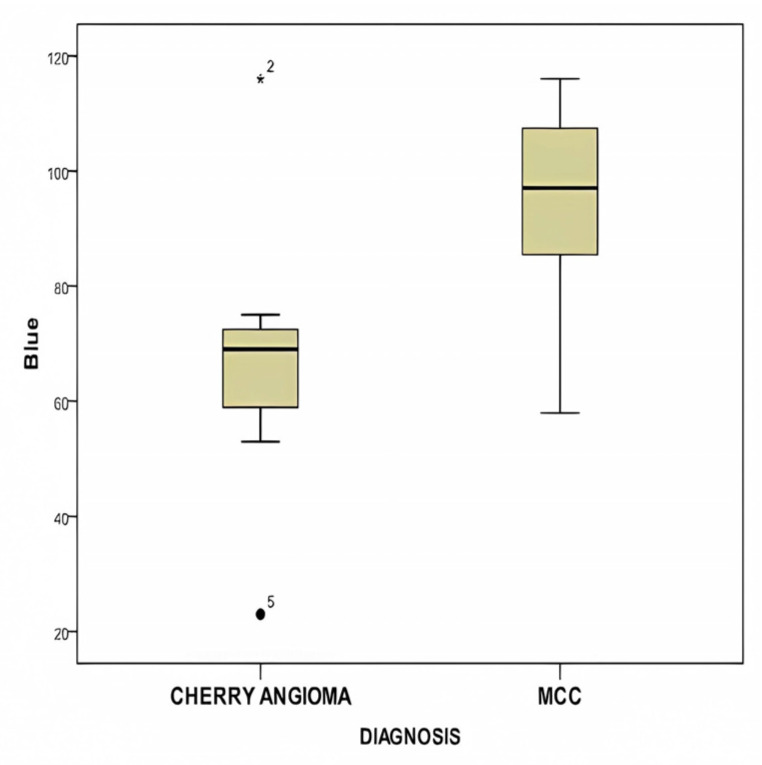
This figure displays the box plot comparing the mean blue color values of RGB between MCC and cherry hemangioma, showing a statistically significant difference (*p* = 0.03).

**Figure 7 diagnostics-14-00230-f007:**
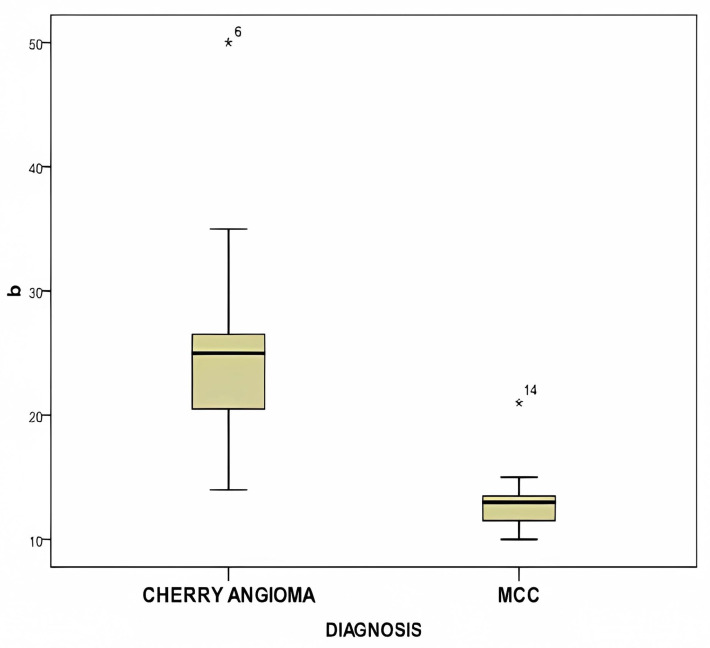
This figure displays the box plot comparing the mean b* color values between MCC and cherry hemangioma, showing a statistically significant difference (*p* < 0.0001).

**Figure 8 diagnostics-14-00230-f008:**
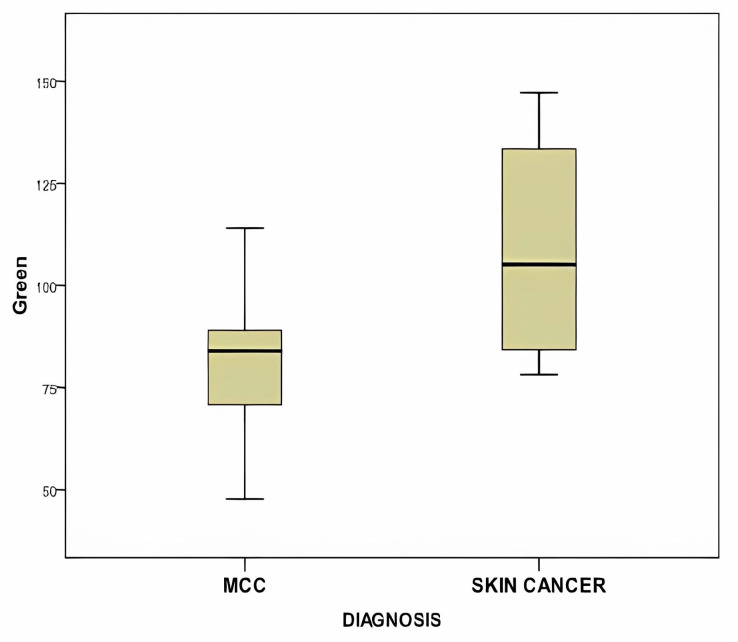
This figure displays the box plot comparing the mean green color value of RGB between MCC, BCC, and SCC, showing a statistically significant difference (*p* = 0.03).

**Figure 9 diagnostics-14-00230-f009:**
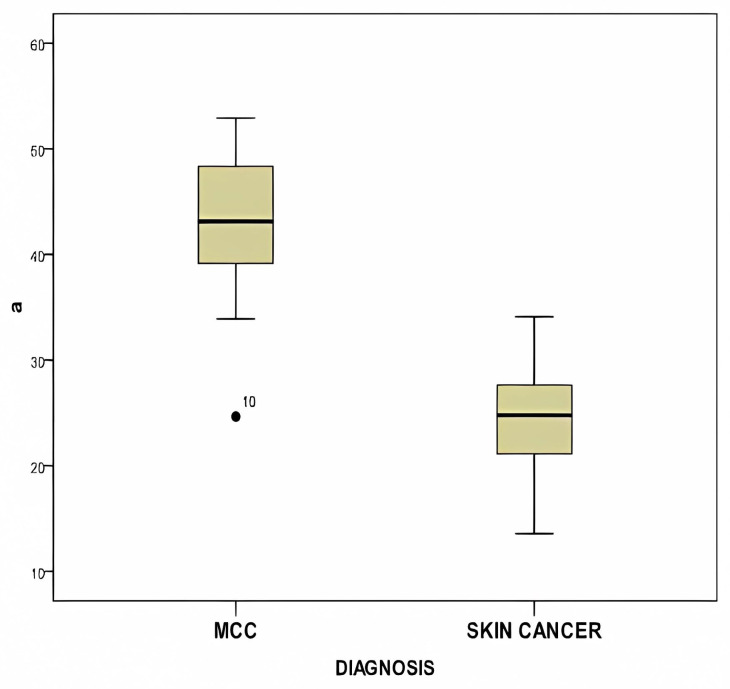
This figure displays the box plot comparing the mean a* value of MCC, BCC, and SCC, showing a statistically significant difference (*p* < 0.0001).

**Figure 10 diagnostics-14-00230-f010:**
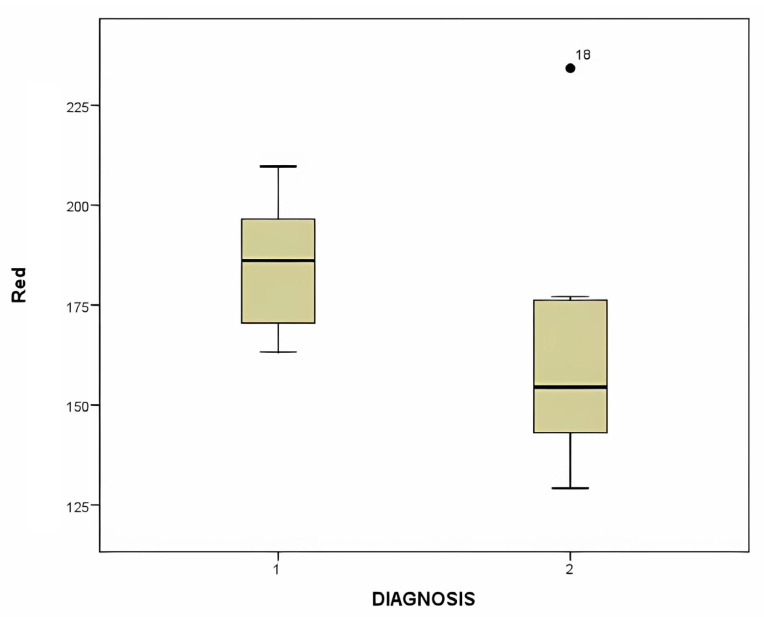
This figure displays the box plot comparing the mean red value of MCC and hemangioma, showing a statistically significant difference (*p* = 0.030).

**Table 1 diagnostics-14-00230-t001:** This table displays the results for red, green, blue (RGB), L*, a*, b*, and ITA across examined lesions, including MCC (number = 11), cherry hemangioma (number = 11), BCC (number = 5), SCC (number = 6), and hemangioma (number = 11).

Patient	Red	Green	Blue	L*	a*	b*	ITA	DIAGNOSIS
1	195.36	61.56	68.57	46.27	53.46	26.45	−8.02	CHERRY
2	234.29	130.99	116.65	65.79	38.13	25.28	31.98	CHERRY
3	180.92	83.31	75.92	47.71	39	23.43	−5.58	CHERRY
4	179.48	69.44	73.92	44.86	45.07	20.86	−13.83	CHERRY
5	149.01	29.35	23.61	32.59	48.2	35.04	−26.42	CHERRY
6	252.19	108.81	64.77	63.35	52.27	50.71	14.75	CHERRY
7	131.08	66.53	53.68	36.09	26.39	20.32	−34.4	CHERRY
8	186.13	54.34	70.36	43.61	53.41	21.63	−16.45	CHERRY
9	145.06	75.35	54.33	40.2	27.35	25.57	−20.96	CHERRY
10	138.91	78.28	72.55	40.32	24.8	14.52	−33.69	CHERRY
11	193.87	71.17	69.33	47.51	48.99	27.5	−5.17	CHERRY
12	188.32	60.31	85.61	45.17	52.88	14.39	−18.56	MCC
13	167.45	73.89	89.13	44.05	39.96	10.13	−30.44	MCC
14	163.26	47.67	58.33	38.22	47.87	21.27	−28.98	MCC
15	202.65	84.05	103.24	51.81	48.76	13.26	7.79	MCC
16	196.32	111.59	116.04	56.62	33.9	12.29	28.33	MCC
17	186.11	89.27	97.74	50.12	39.84	13.8	0.5	MCC
18	171.24	79.15	86.69	45.61	38.44	13.88	−17.55	MCC
19	169.73	67.73	82.08	43.1	43.11	13.22	−27.55	MCC
20	209.73	88.73	112.24	53.96	49.83	11.06	19.69	MCC
21	176.62	114.04	113.53	54.61	24.66	10.55	23.59	MCC
22	196.77	84.82	97.26	50.89	45.82	15.41	3.31	MCC
23	195.47	120.94	105.15	58.37	27.35	20.94	21.78	BCC/SCC
24	205.99	147.21	133.26	66.33	20.35	16.19	45.25	BCC/SCC
25	164.11	80.72	77.92	44.66	34.09	17.72	−16.78	BCC/SCC
26	187.8	110.03	107.45	54.97	30.52	14.81	18.54	BCC/SCC
27	196.88	127.15	133.94	60.6	27.89	7.59	54.4	BCC/SCC
28	149.7	83.91	79.4	43.29	26.77	14.62	−24.65	BCC/SCC
29	141.82	85.64	87.17	42.8	23.5	9.04	−38.56	BCC/SCC
30	114.46	78.18	65.55	36.88	13.57	13.61	−43.96	BCC/SCC
31	145.93	100.11	92.25	47	17.46	11.91	−14.15	BCC/SCC
32	146.09	84.64	68.31	42.73	23.73	20.49	−19.53	BCC/SCC
33	221.22	146.6	126.2	67.77	25.81	22.37	38.47	BCC/SCC
34	198.8	139.69	135.18	63.78	21.91	11.43	50.33	BCC/SCC
35	117.62	43.26	50.63	28.92	33.27	12.43	−59.48	HEMANGIOMA
36	144.72	81.6	92.8	42.39	27.69	5.06	−56.38	HEMANGIOMA
37	134.11	80.22	82.49	40.35	22.89	8.32	−49.22	HEMANGIOMA
38	166.23	81.71	90.37	45.45	35.61	11.31	−21.91	HEMANGIOMA
39	152.46	83.97	98.08	44.15	30.06	4.48	−52.59	HEMANGIOMA
40	139.55	49.53	70.75	34.46	40.06	7.55	−64.09	HEMANGIOMA
41	231.25	113.34	123.18	61.96	46.52	16.32	36.23	HEMANGIOMA
42	147.37	80.24	93.73	42.48	29.61	4.67	−58.17	HEMANGIOMA
43	157.88	74.23	74.29	42.28	34.78	16.58	−24.96	HEMANGIOMA
44	136.11	45.77	52.48	32.77	39	16.94	−45.48	HEMANGIOMA
45	178.88	98.04	86.68	50.68	31.4	20.94	1.86	HEMANGIOMA
46	147.18	49.5	44.21	35.28	40.74	25.73	−29.76	HEMANGIOMA

**Table 2 diagnostics-14-00230-t002:** Comparison of mean values for red, green, blue (RGB), L*, a*, b*, and ITA across examined lesions of MCC, cherry hemangioma, BCC, SCC, and hemangioma.

	Red	Green	Blue	L*	a*	b*	ITA
CHERRY	180.57	75.38	67.61	46.21	41.55	26.48	−10.71
MCC	184.38	81.93	94.72	48.56	42.28	13.57	−3.62
BCC and SCC	172.36	108.73	100.98	52.43	24.41	15.06	5.93
HEMANGIOMA	154.45	73.45	79.97	41.76	34.3	12.53	−35.33

**Table 3 diagnostics-14-00230-t003:** Statistical comparisons of red, green, blue (RGB), L*, a*, b*, and ITA across examined lesions comparing MCC with cherry hemangioma. Significantly different values were observed in the blue color of RGB (*p* = 0.003) and the b* parameter (*p* < 0.0001).

	Red	Green	Blue	L*	a*	b*	ITA
Mann–Whitney U	51.5	44	17.5	43	56	5	50
Wilcoxon W	117.5	110	83.5	109	122	71	116
Z	−0.591	−1.084	−2.825	−1.151	−0.297	−3.66	−0.69
Asymp. Sig. (2-tailed)	0.554	0.278	0.005	0.25	0.767	0	0.49

**Table 4 diagnostics-14-00230-t004:** Statistical analysis comparing MCC with BCC and SCC. Significantly different values were observed in the green color of RGB (*p* = 0.037) and the a* parameter (*p* < 0.0001).

	Red	Green	Blue	L*	a*	b*	ITA
Mann–Whitney U	53	32	59	55	7	51	55
Wilcoxon W	131	98	125	121	85	117	121
Z	−0.8	−2.093	−0.431	−0.677	−3.631	−0.923	−0.677
Asymp. Sig. (2-tailed)	0.424	0.036	0.667	0.498	0	0.356	0.498
Exact Sig. (2*(1-tailed Sig.))	0.449	0.037a	0.695	0.525	0.000	0.379	0.525

**Table 5 diagnostics-14-00230-t005:** Statistical analysis comparing MCC with hemangioma. Significant differences were found in the red color of RGB (*p* = 0.002), L* parameter (*p* = 0.019), a* parameter (*p* = 0.023), and ITA (*p* = 0.009).

	Red	Green	Blue	L*	a*	b*	ITA
Mann–Whitney U	17	50	40	28	29	58	24
Wilcoxon W	95	128	118	106	107	136	102
Z	−3.016	−0.985	−1.6	−2.339	−2.277	−0.492	−2.585
Asymp. Sig. (2-tailed)	0.003	0.325	0.11	0.019	0.023	0.622	0.01
Exact Sig. (2*(1-tailed Sig.))	0.002	0.347	0.118	0.019	0.023	0.651	0.009

## Data Availability

The data that support the findings of this study are available upon reasonable request from the corresponding author, D.K.

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
