# Peer review of "Color Analysis of Merkel Cell Carcinoma: A Comparative Study with Cherry Angiomas, Hemangiomas, Basal Cell Carcinomas, and Squamous Cell Carcinomas"

_diagnostics, 2024, doi:10.3390/diagnostics14020230_

Round 1

Reviewer 1 Report (New Reviewer)

Comments and Suggestions for Authors

Dear Authors, 

 Regarding the article I read, I have the following to point out:

The editing rules according to the magazine's requirements are not followed (introduction of figures in the text)

In paragraph 184 the abbreviation for melanoma is made again

There is text that is repeated in this article:  the paragraphs 243-247  and 250-255 are the same!!

The tables do not respect the drafting rules

The abbreviation MCC was inserted in the text at least 4 times

NO bibliographical reference is written correctly

Author Response

Reply to reviewer 1

Comments and Suggestions for Authors

Dear Authors, 

 Regarding the article I read, I have the following to point out:

The editing rules according to the magazine's requirements are not followed (introduction of figures in the text)

Reply

We have now followed the editing rules according to the magazine's requirements.

In paragraph 184 the abbreviation for melanoma is made again

Reply

In paragraph 184 we have not repeated now the abbreviation for melanoma

There is text that is repeated in this article:  the paragraphs 243-247  and 250-255 are the same!!

Reply

The tables do not respect the drafting rules

Reply

Now the tables respect the drafting rules

The abbreviation MCC was inserted in the text at least 4 times

Reply

Now, the abbreviation MCC was inserted in the text one time

NO bibliographical reference is written correctly

Reply

We have now corrected the bibliographical reference

Reviewer 2 Report (New Reviewer)

Comments and Suggestions for Authors

Dec 18, 2023

Dear Author,

1.     The manuscript, diagnostics-2755036, is within the scope of the journal.

2.     The section “Abstract” should be revised.

3.     The section “Keywords” must be compatible with MeSH database recommendations.

4.     In my opinion, Basal Cell Carcinoma is a critical issue, particularly in Dermatopathology. To this end we have also emphasized its cruciality, to date. So, we evaluate the author’s hypothesis as valuable. As such, the section “Discussion” must be revised and enriched by discussing this essential issue, and the reference list must be enriched with several articles, including “CD 10 for the distinct differential diagnosis of basal cell carcinoma and benign tumours of cutaneous appendages originating from hair follicle”, “Differential diagnosis of basal cell carcinoma and benign tumors of cutaneous appendages originating from hair follicles by using CD34”, and “Histopathological characteristics may not be useful in the differential diagnosis between basal cell carcinoma and benign tumors of cutaneous appandages originating from hair follicle.”

5.     The section “References” must be compatible with author guidelines of the journal, Diagnostics.

6.     The figures must be revised by some contrast-enhancing modalities.

7.     The orthographical and grammatical errors should be revised by the authors.

8.     The manuscript might be accepted for publication in Diagnostics after minor revision.

Best Regards,

Reviewer, Diagnostics

Comments on the Quality of English Language

The orthographical and grammatical errors should be revised by the authors.

Author Response

Reply to reviewer 2

Dear Author,

  1. The manuscript, diagnostics-2755036, is within the scope of the journal.
  2. The section “Abstract” should be revised.

Reply

Now the section “Abstract” has been revised

Abstract: Merkel cell carcinoma (MCC) is recognized as one of the most malignant skin tumors. Its rarity might explain the limited exploration of digital color studies in this area. The objective of this study was to delineate color alterations in MCCs compared to benign lesions resembling MCC, such as cherry angiomas and hemangiomas, along with other non-melanoma skin cancer lesions like basal cell carcinoma (BCC) and squamous cell carcinoma (SCC), utilizing computer-aided digital color analysis. This was a retrospective study where clinical images of the color of the lesion and adjacent normal skin from 11 patients with primary MCC, 11 patients with cherry angiomas, 12 patients with hemangiomas, and 12 patients with BCC/SCC (totaling 46 patients) were analyzed using the RGB (red, green, and blue) and the CIE Lab color system. The Lab color system aided in estimating the Individual Typology Angle (ITA) change in the skin, and these results are documented in this study. It was demonstrated that the estimation of color components can assist in the differential diagnosis of these types of lesions since there were significant differences in color parameters between MCC and other categories of skin lesions such as hemangiomas, common skin carcinomas, and cherry hemangiomas. Significant differences in values were observed in the blue color of RGB (p = 0.003) and the b* parameter of Lab color (p < 0.0001) of MCC versus cherry angiomas. Similarly, the mean a* value of Merkel cell carcinoma (MCC) compared to basal cell carcinoma and squamous cell carcinoma showed a statistically significant difference (p < 0.0001). Larger prospective studies are warranted to further validate the clinical application of these findings.

  1. The section “Keywords” must be compatible with MeSH database recommendations.

Reply

Now, the section “Keywords” is compatible with MeSH database recommendations

Keywords: Merkel cell carcinoma (MCC); neuroendocrine carcinoma; skin cancer, basal cell carcinoma, squamous cell carcinoma, cherry angiomas, color analysis, image processing, Computer-aided Diagnostics; Digital Dermatology

  1. In my opinion, Basal Cell Carcinoma is a critical issue, particularly in Dermatopathology. To this end we have also emphasized its cruciality, to date. So, we evaluate the author’s hypothesis as valuable. As such, the section “Discussion” must be revised and enriched by discussing this essential issue, and the reference list must be enriched with several articles, including “CD 10 for the distinct differential diagnosis of basal cell carcinoma and benign tumours of cutaneous appendages originating from hair follicle”, “Differential diagnosis of basal cell carcinoma and benign tumors of cutaneous appendages originating from hair follicles by using CD34”, and “Histopathological characteristics may not be useful in the differential diagnosis between basal cell carcinoma and benign tumors of cutaneous appendages originating from hair follicle.”

Reply

Now the reference list has been enriched with these articles.

Several studies have sought to identify specific markers for differentiating skin cancer from other benign tumors. One study aimed to address the challenges in differentiating between trichoadenoma, trichofolliculoma, trichoepithelioma, trichoblastoma, and BCC, especially with small specimens. The researchers examined 30 cases each of benign tumors from hair follicle appendages and BCC. CD10 expression was assessed in both tumor groups, and results indicated stronger stromal CD10 immunopositivity in benign tumors, while peripheral CD10 immunopositivity was stronger in BCC. The study suggests that CD10 expression analysis can be a valuable tool for differential diagnosis, especially in small and superficial biopsies, potentially impacting treatment decisions in selected cases. Another study aimed to address challenges in differentiating between benign tumors of cutaneous appendages originating from hair follicles (BTCOHF) and BCCs, particularly with small biopsy specimens. The researchers investigated the potential utility of CD34 expression for assistance in this differential diagnosis. Results showed that [1+] and [2+] immunopositivity of CD34 in BTCOHFs was significantly stronger than in BCCs, suggesting that CD34 expression analysis may contribute to the differential diagnosis of these skin lesions.

Sengul D, Sengul I, Astarci MH, Ustun H, Mocan G. CD10 for the distinct differential diagnosis of basal cell carcinoma and benign tumours of cutaneous appendages originating from hair follicle.Pol J Pathol. 2010;61(3):140-6.PMID: 21225496 

Sengul D, Sengul I, Astarci MH, Ustun H, Mocan G. Differential diagnosis of basal cell carcinoma and benign tumors of cutaneous appendages originating from hair follicles by using CD34. Asian Pac J Cancer Prev. 2010;11(6):1615-9.PMID: 21338206

Astarci HM, Unsal G, Sengul D, Hucumenoglu S, Kocer U, Ustun H. Significance of androgen receptor and CD10 expression in cutaneous basal cell carcinoma and trichoepithelioma.

Oncol Lett. 2015 Dec;10(6):3466-3470. doi: 10.3892/ol.2015.3804. Epub 2015 Oct 13.PMID: 26788151 

  1. The section “References” must be compatible with the author guidelines of the journal, Diagnostics.

Reply

Now, the section “References” is compatible with the author guidelines of the journal, Diagnostics

  1. The figures must be revised by some contrast-enhancing modalities.

Reply

Now the figures have been revised by some contrast-enhancing modalities. We also include all figures in one pdf file.

  1. The orthographical and grammatical errors should be revised by the authors.

Reply

Now, the orthographical and grammatical errors have been revised.

  1. The manuscript might be accepted for publication in Diagnosticsafter minor revision.

Reply

We have elaborated these revisions and we provide the revised manuscript.

Best Regards,

Reviewer, Diagnostics

Comments on the Quality of English Language

The orthographical and grammatical errors should be revised by the authors.

Submission Date

19 November 2023

Date of this review

18 Dec 2023 13:11:10

Reviewer 3 Report (New Reviewer)

Comments and Suggestions for Authors

The authors reported the results of a study with the aim of delineating color alterations in MCCs compared to benign lesions resembling MCC, such as cherry angiomas and hemangiomas, along with other non-melanoma skin cancer lesions like basal cell carcinoma (BCC) and squamous cell carcinoma (SCC), utilizing computer-aided digital color analysis. The manuscript is interesting ad well written. The topic is original and relevant in the field as updated data are needed. Moreover, the study methods improved the relevance of the results, offering a wide perspective. Strengths and limitations of the study have been discussed. References are appropriate. Tables and figures improve the quality of the paper. The manuscript is suitable for publication.

Author Response

Reviewer 3

Open Review

The authors reported the results of a study with the aim of delineating color alterations in MCCs compared to benign lesions resembling MCC, such as cherry angiomas and hemangiomas, along with other non-melanoma skin cancer lesions like basal cell carcinoma (BCC) and squamous cell carcinoma (SCC), utilizing computer-aided digital color analysis. The manuscript is interesting ad well written. The topic is original and relevant in the field as updated data are needed. Moreover, the study methods improved the relevance of the results, offering a wide perspective. Strengths and limitations of the study have been discussed. References are appropriate. Tables and figures improve the quality of the paper. The manuscript is suitable for publication.

Reply

Dear reviewer,

Thank you for your comments.

This manuscript is a resubmission of an earlier submission. The following is a list of the peer review reports and author responses from that submission.

Round 1

Reviewer 1 Report

Comments and Suggestions for Authors

Koumaki et al. conducted a color analysis of MCC in 11 cases using RGB and the CIE Lab color model for digital patient images.

Major comment:

·         While this represents the first study to employ a digital algorithm for analyzing MCC, the results simply affirm that MCC appears "redder" than normal skin, a well-known clinical feature that can be readily discerned with the naked eye. The clinical significance of these findings remains unclear. It's essential to elucidate how this technique can enhance the specificity of MCC diagnosis and distinguish it from other benign tumors. If such improvements exist, please present relevant data to support this claim.

Minor comments:

·         In the Introduction, consider streamlining the content. The discussion of etiology and treatment updates may not be necessary. Instead, emphasize the non-specific nature of MCC and the diagnostic challenges it presents if you intend to argue for enhanced diagnostic specificity using color analysis (as mentioned in the major comment above). Consider referencing articles that address MCC's clinical non-specificity.

·         Explain why the authors specifically chose RGB and the CIE Lab color model for MCC analysis.

·         Ensure that the manuscript addresses how the digital image analysis was adjusted to account for different lighting conditions, device variables, and other relevant factors.

Comments on the Quality of English Language

None

Reviewer 2 Report

Comments and Suggestions for Authors

Authors tried to observe the changes of colors in MCCs compared to the surrounding normal skin regarding digital color analysis for accurate diagnosis. The technical aspect of used techniques could be of some interests. However, the general quality of the manuscript should be improved significantly.

  1. The introduction part should explain why using color technique can be justified. The explanation of MCCs or dermoscopic findings are not strongly relevant. They should explain what is the used technique and rationale to use it for potential better diagnosis.

  2. Materials and methods should describe how many cases were assessed for inclusion and how many cases were excluded for what kind of reasons and how many cases were finally included.

  3. Most of the tables should be made into figures, otherwise readers would not understand the results.

  4. The current manuscript entirely failed to provide any evidence on the diagnostic ability of the current technique for better diagnosis compared to currently-used-methods such as dermoscopy.

  5. The authors described “what becomes evident when comparing the color changes is a notable decline in the presence of green in the tumor regions, accompanied by a decrease in the ITA angle.”. However, this is not surprising at all because MCCs are known to be a reddish tumor (consistent with the clinical photo in the current manuscript).  Authors should show the advantage of using this color measurement for better diagnosis.

Comments on the Quality of English Language

Should be improved to make it more concise.
